# Lung macrophages utilize unique cathepsin K–dependent phagosomal machinery to degrade intracellular collagen

Ivo Fabrik[1,2,3] , Orsolya Bilkei-Gorzo[1,2,*], Maria Öberg[1,2,*], Daniela Fabrikova[1,2], Johannes Fuchs[4], Carina Sihlbom[4], Melker Göransson[5] , Anetta Härtlova[1,2,6]

Resident tissue macrophages are organ-specialized phagocytes responsible for the maintenance and protection of tissue homeostasis. It is well established that tissue diversity is reflected by the heterogeneity of resident tissue macrophage origin and phenotype. However, much less is known about tissue-specific phagocytic and proteolytic macrophage functions. Here, using a quantitative proteomics approach, we identify cathepsins as key determinants of phagosome maturation in primary peritoneum-, lung-, and brain-resident macrophages. The data further uncover cathepsin K (CtsK) as a molecular marker for lung phagosomes required for intracellular protein and collagen degradation. Pharmacological blockade of CtsK activity diminished phagosomal proteolysis and collagenolysis in lung-resident macrophages. Furthermore, profibrotic TGF-$\beta$ negatively regulated CtsK-mediated phagosomal collagen degradation independently from classical endocytic–proteolytic pathways. In humans, phagosomal CtsK activity was reduced in COPD lung macrophages and non-COPD lung macrophages exposed to cigarette smoke extract. Taken together, this study provides a comprehensive map of how peritoneal, lung, and brain tissue environment shapes phagosomal composition, revealing CtsK as a key molecular determinant of lung phagosomes contributing to phagocytic collagen clearance in lungs.

## Introduction

Resident tissue macrophages (RTMs) are present in every organ of our body and play an essential role in host protection against infection (Bleriot et al, 2020). As professional phagocytes, they are specialized in the recognition, engulfment, and digestion of not only pathogens but also apoptotic cell debris (Murray & Wynn, 2011).

Thus, the phagocytic process is essential not only for microbial elimination, but also for the maintenance of tissue homeostasis and remodelling (Rothlin et al, 2007; Lemke, 2013; Arandjelovic & Ravichandran, 2015).

Phagocytosis is a highly conserved process characterized by recognition and ingestion of particles larger than 0.5 $\mu m$ into a plasma membrane–derived vesicle, known as phagosome (Pauwels et al, 2017). After internalization, phagosomes undergo a series of fusion steps with endosomal compartment and ultimately with lysosome to form a highly acidic phagolysosome. During this last step, the phagolysosome acquires important components for the degradation of particle, including proteases, nucleases, and lipases required for digestion of engulfed material (Desjardins et al, 1994; Fairn & Grinstein, 2012). This process needs to be tightly regulated for phagocytes to carry out their function in immunity and homeostasis. If uncontrolled, it can lead to the development of persistent infection or autoimmunity and inflammatory disorders (Johnson & Newby, 2009; Nagata et al, 2010; Colegio et al, 2014).

It is now well established that most of the RTMs are derived from embryonic yolk sac and foetal liver precursors independently of the contribution of bone marrow precursors (Ginhoux & Guilliams, 2016; Bleriot et al, 2020). As such, specific macrophage responses to a phagocytic prey are defined not only by the use of phagocytic receptor and activation of distinct signalling pathways but also by microenvironment-dependent signals. However, the mechanisms by which tissue milieu impacts macrophage phagocytic capacity and downstream phagosomal processing are poorly understood. We hypothesized that different tissue environment will modulate the phagosomal system of distinct RTMs and will contribute to their organ-specific functions.

Here, by employing high-resolution mass spectrometry–based proteomics, we describe differences in the phagosomal environment of primary macrophages from distinct tissues. Our data reveal an essential role of cathepsin proteases as drivers of RTM-specific phagosomal proteolytic activity. Specifically, we identify cathepsin K as

[1]Institute of Biomedicine, Department of Microbiology, The Sahlgrenska Academy, University of Gothenburg, Gothenburg, Sweden   [2]Wallenberg Centre for Molecular and Translational Medicine, University of Gothenburg, Gothenburg, Sweden   [3]Biomedical Research Centre, University Hospital Hradec Kralove, Hradec Kralove, Czech Republic   [4]Proteomics Core Facility, The Sahlgrenska Academy, University of Gothenburg, Gothenburg, Sweden   [5]Bioscience COPD/IPF, Research and Early Development, Respiratory and Immunology (R&I), BioPharmaceuticals R&D, AstraZeneca, Gothenburg, Sweden   [6]Institute of Medical Microbiology and Hygiene, Faculty of Medicine, Medical Center—University of Freiburg, Freiburg, Germany

Correspondence: anetta.hartlova@gu.se; anetta.hartlova@uniklinik-freiburg.de; ivo.fabrik@fnhk.cz
*Orsolya Bilkei-Gorzo and Maria Öberg contributed equally to this work

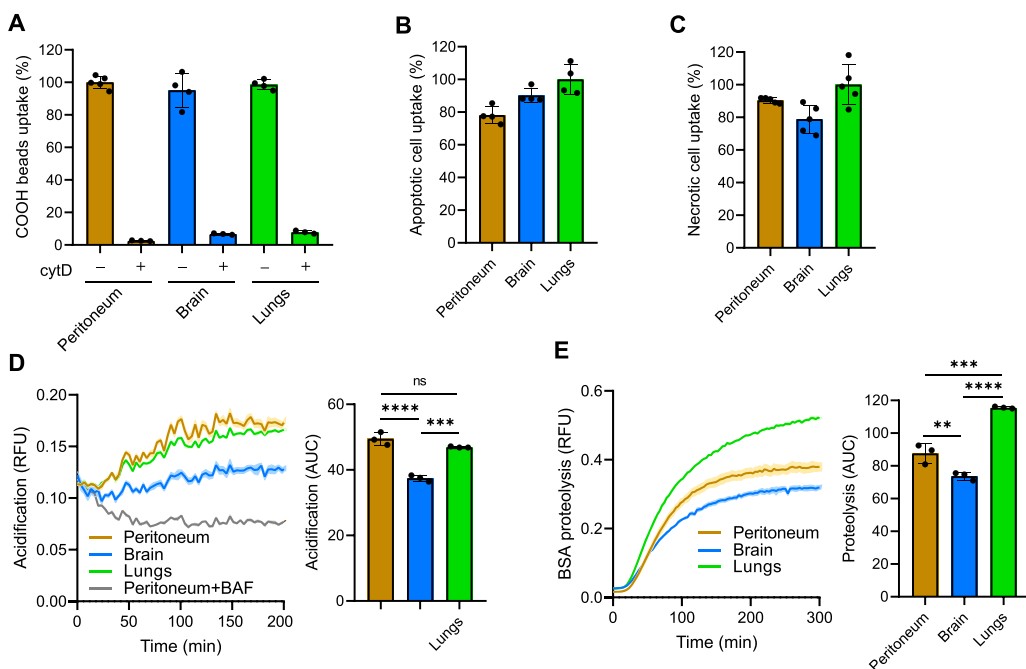

**Figure 1. Phagosomes of RTMs mature at different rates.**

**(A, B, C)** Uptake of BSA-coated negatively charged silica particles (A), apoptotic cells (B), and necrotic cells (C) by RTMs isolated from peritoneum, brain, and lungs. Inhibition of phagocytosis by 10 μM cytochalasin D was done 1 h before and during the experiment. **(D, E)** Kinetic measurement of phagosomal acidification (D) and proteolysis (E) in peritoneal, brain, and lung RTMs exposed to BSA-coated particles. Inhibition of phagosomal acidification by 100 nM bafilomycin A1 was done 1 h before and during the experiment. Data information: the statistical significance of data is denoted on graphs by asterisks where **$P < 0.01$, ***$P < 0.001$, ****$P < 0.0001$, or ns, not significant as determined by ANOVA with the post hoc Tukey test. Data in bar graphs (A, B, C) are shown as means of normalized fluorescence ± SD. Data showing time-dependent fluorescence (D, E) are normalized to bead uptake (relative fluorescence units, RFUs) and expressed as means of RFU ± SEM or as area under the curve ± SD. Data are representative of two (D) or three (A, B, C, E) replicates.

a phagosomal marker of mouse and human lung macrophages with a crucial function in phagosomal proteolysis and collagen degradation. Importantly, cathepsin K–mediated phagosomal collagenolysis in lung RTMs is negatively regulated by fibrosis-related stimuli and reduced in macrophages from chronic obstructive pulmonary disease (COPD) patients. Altogether, these data uncover that phagosomal distribution of cathepsins within distinct RTMs reflects their local substrates and determines their tissue-specific functions.

# Results and Discussion

### RTM phagosomes mature at a distinct rate

We first investigated whether tissue residency impacts macrophage phagocytic activity. RTMs isolated for phagocytic profiling included brain, lung, and peritoneal macrophages. These three different RTMs represent cells of distinct macrophage origin with a specialized function essential for their tissue environment. While brain macrophages are mainly homeostatic cells providing the main cellular defence of the central nervous system, lung macrophages maintain lung physiology by clearance of pulmonary surfactant and elimination of microorganisms or allergens (Guilliams et al, 2013; Li & Barres, 2018). Peritoneal macrophages are mainly important for protection against pathogens (Cassado et al, 2015). Distinct macrophage populations were isolated as described in

the Materials and Methods section, and their phenotypes were confirmed by flow cytometry (Fig S1). Briefly, peritoneal macrophages were distinguished by CD11b⁺ F4/80⁺, brain macrophages were isolated by positive-binding enrichment to CD11b magnetic beads, and lung macrophages were composed of two macrophage subsets: interstitial (CD11c⁺, CD11b⁺, F4/80⁺, Siglec-F–negative) and alveolar (CD11c⁺, CD11b⁺ low, F4/80⁺, Siglec-F–positive) lung macrophages (Draijer et al, 2019). To determine phagocytic activity of different RTMs, we examined the uptake of carboxylated beads, which serve as surrogates to apoptotic cells in RTMs cultivated or not with L929 cell-conditioned media (LCCM) for 3 d (Figs 1A and S2A) (Guo et al, 2019; Bilkei-Gorzo et al, 2022). There were no phenotypic differences between different RTMs of carboxylated bead uptake with or without LCCM, but LCCM improved RTM adherence and rate of phagocytosis. Next, we tested the uptake of apoptotic and necrotic cells by RTMs in the presence of LCCM. Apoptosis was induced by treatment with cycloheximide, whereas necrotic cells were induced by repeated freeze/thaw cycles (Fig 1B and C) (Martinez et al, 2011). Our analysis did not reveal any notable differences in uptake among RTMs, despite previous report describing distinct phagocytic pathways preferred by different RTMs (A-Gonzalez et al, 2017). Overall, these data suggest similar rates of cell debris removal by different RTMs under homeostatic conditions, probably because of functional redundancy of RTM-specific phagocytic components.

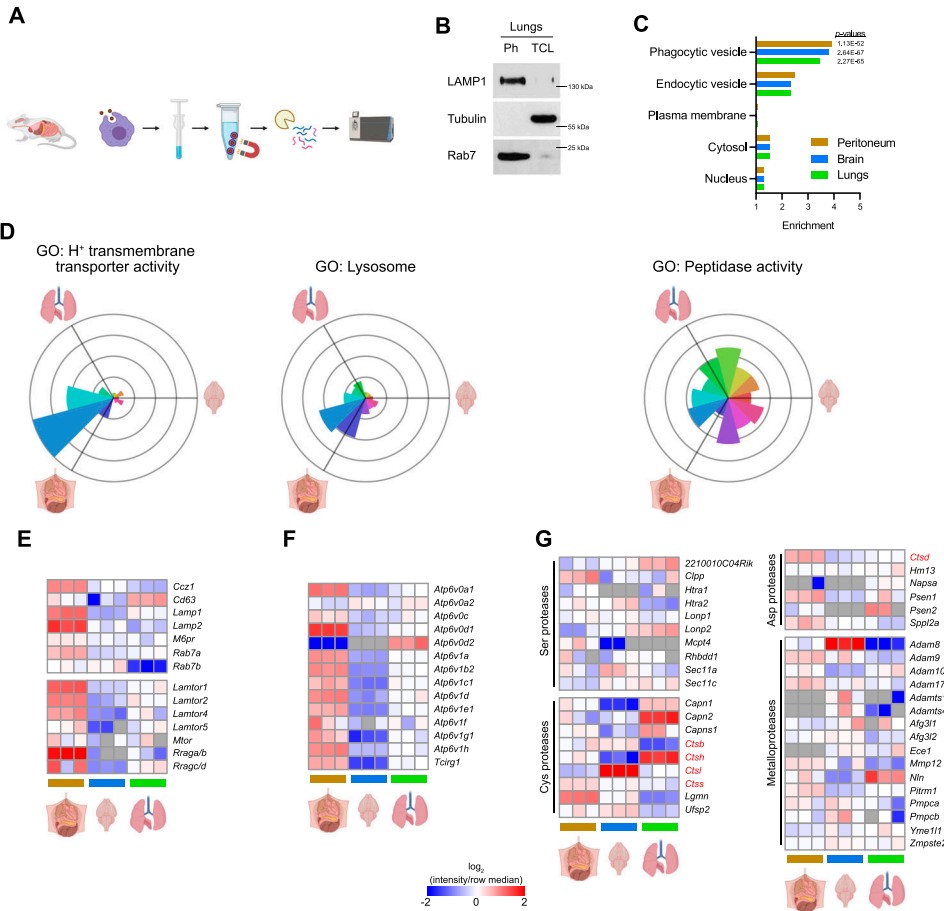

**Figure 2. Proteins involved in phagosome–lysosome fusion are not the key determinants of phagosomal proteolytic activity.**
**(A)** Workflow for isolation of phagosomes from peritoneal, brain, and lung RTMs for proteomics. **(B)** Purity of isolated phagosomes as determined by Western blot. The presence of phagosomal markers LAMP1 and Rab7 was probed in phagosome samples (Ph) and corresponding total cell lysates. Samples from lung RTMs are shown for demonstration. **(C)** Enrichment of phagosome-related Gene Ontology Cellular Component terms in RTM phagosome dataset. **(D)** Relative distribution of proteins connected to phagosome maturation across phagosomes of peritoneal, brain, and lung RTMs. The radius of each sector in rose plot corresponds to a relative up-regulation of associated proteins in the direction of the given tissue. **(E, F, G)** Heatmaps showing relative expression of components of lysosomal markers (upper part) and proteins involved in mTORC1 signalling from phagosome (E), vacuolar ATPase (F), and endopeptidases (G) in phagosomes of peritoneal, brain, and lung RTMs. Colour gradient correlates to the log-ratio of intensity divided by the row median; grey cells indicate missing values. Data information: Western blots (B) are representative of three replicates. Proteomics data (C, D, E, F, G) are derived from three replicates. Enrichment of GO terms (C) was determined by the Fisher exact test yielding highlighted *P*-values for selected terms. Data in rose plots (D) are averaged from at least two replicates. Illustrations were created with BioRender.com.

Next, we decided to determine whether phagosome maturation varies in distinct RTMs by analysing phagosomal pH and proteolysis in real time (Yates et al, 2007; Podinovskaia et al, 2013). While peritoneal and lung macrophages exhibited comparable rates and extents of phagosomal acidification, brain macrophages profoundly showed a smaller proportion of acidified phagosomes indicating less degradative capacity of brain phagosomes compared with lung and peritoneum independent of LCCM (Figs 1D and S2B and C). Notably, the proteolytic efficiency of lung phagosomes was significantly enhanced compared with peritoneal macrophages in spite of a similar rate of phagosomal acidification. These data could be explained by a faster fusion of lung phagosomes with lysosomes and/or by a higher concentration of proteases in lysosomes of lung macrophages compared with peritoneal macrophages independent of LCCM (Figs 1E and S2D and E). Taken together, these data suggest that phagosome maturation contributes to functional heterogeneity of RTMs.

## RTM phagosomal proteome

To better understand organ-specific phagosome maturation of macrophages in more detail, we further investigated which of the phagosome-associated molecules regulate the proteolytic activity

of phagosomes across different tissues. Following phagocytic functional assays and due to demands of proteomics approach, we decided to cultivate all mouse RTMs in the presence of LCCM for 3 d; LCCM was washed away the day before induction of phagocytosis. Phagosomes of peritoneum-, lung-, and brain-resident macrophages were isolated using magnetic beads, and phagosomal proteomes were analysed by label-free quantitative tandem mass spectrometry (Fig 2A and Table S1). Phagosome purity was confirmed by immunoblot showing enrichment of a prototypical phagosomal marker, Rab7, and phagosome–lysosome marker, LAMP1, within phagosomal fraction while cytosolic protein, tubulin, was absent (Fig 2B). In line with the immunoblot data, Gene Ontology enrichment analysis of proteomics data confirmed enrichment of proteins involved in phagocytosis and endocytosis in the phagosomal fraction compared with low abundant cytosolic and nuclear contaminants (Fig 2C).

Phagosome–lysosome fusion is a stepwise process in which phagocytic cells drive the maturation of nascent phagosomes to highly degradative, mature phagolysosomes. During this transition, the lumens of maturing phagosomes undergo a transformation characterized by decreasing pH, changes in ion composition, and the addition of various hydrolases. RTMs displayed a marked variety in the composition of phagosomal proteome (Fig S3A). While

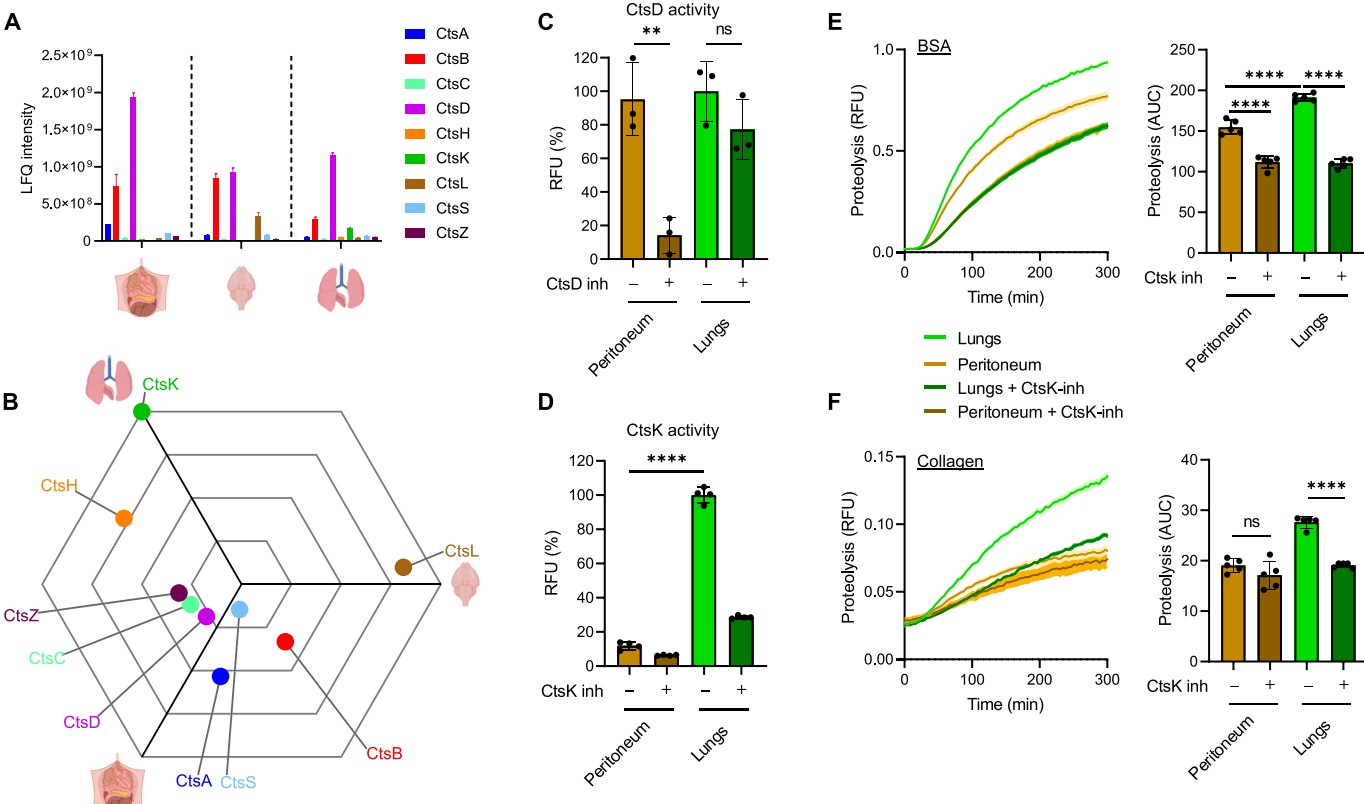

**Figure 3. Cathepsin K enriched in phagosomes of lung RTMs is responsible for phagosomal degradation of collagen.**
**(A)** Normalized LFQ intensities of individual cathepsins in RTM phagosomes as determined by MS. **(B)** Relative distribution of cathepsins in RTM phagosomes. Distance from the centre of the triwise plot corresponds to a relative up-regulation of cathepsins in the direction of the given tissue. **(C, D)** Activity of cathepsin D (C) and cathepsin K (D) in peritoneal and lung RTMs. Measured from lysates; pepstatin A (CtsD inh) and L 006235 (CtsK inh) were used at $1\,\mu M$ concentrations, and were added directly into lysates. **(E, F)** Kinetic measurement of phagosomal BSA (E) and collagen (F) degradation in peritoneal and lung RTMs treated with CtsK inhibitor. Inhibition of CtsK by $1\,\mu M$ L 006235 (CtsK inh) was done 1 h before and during the experiment. Data information: the statistical significance of data is denoted on graphs by asterisks where $**P < 0.01$, $****P < 0.0001$, or ns, not significant as determined by ANOVA with the post hoc Tukey test. Data in (A, B) are averaged from at least two replicates. Data in bar graphs (C, D) are shown as means of normalized fluorescence ± SD. Data showing time-dependent fluorescence (E, F) are normalized to the uptake of coated beads and are expressed as means of RFU ± SEM or as area under the curve ± SD. Data are representative of two (C, E) or three (D, F) replicates. Illustrations were created with BioRender.com.

peritoneal macrophages showed significant enrichment of proteins involved in phagosome maturation, lung macrophages displayed enrichment of proteins involved in degradation of ECM and brain macrophages in interferon signalling.

Peritoneal macrophages exhibited the highest number of up-regulated proteins associated with phagosome–lysosome fusion (Figs 2D and S3B), including Rab7a GTPase, LAMP1, and LAMP2 (Fig 2E). In addition, the levels of four members of the LAMTOR complex, a key player in lysosomal trafficking, were significantly enriched in peritoneal phagosomes compared with lung and brain phagosomes (Fig 2E). Though peritoneal and lung macrophages display a similar rate of acidification, peritoneal macrophages showed increased abundance of eight V1 and three catalytic vATPase subunits, which are essential for phagosomal acidification (Fig 2F). These data suggest that the classical phagosome maturation pathway is more active in peritoneal macrophages compared with lung and brain macrophages. Interestingly, all RTMs displayed heterogeneous expression of phagosomal proteolytic enzymes independently of proteins mediating phagosome–lysosome fusion

(Fig 2D). These included aspartic, cysteine, and serine proteases, and metalloproteases such as calpains 1 and 2, legumain, ADAM8, or ADAMTS1/4 with distinct distribution in RTM phagosomes (Fig 2G). Of these, cathepsins showed the most consistent RTM-specific expression (Fig 2G).

## Cathepsins are key determinants of functional specialization of RTMs

Cathepsins are the most abundant proteases of the lysosomal system responsible for degradation of engulfed protein material (Bird et al, 2009; Turk et al, 2012). They are synthesized as inactive proenzymes and can be classified based on their catalytic site into three groups—aspartic, serine, and cysteine proteases. However, their functional relevance to distinct RTMs is still not well understood. Most of the identified phagosomal cathepsins belong to cysteine proteases, whereas the most abundant and ubiquitously present in all three different phagosomes was the best studied aspartic protease, cathepsin D (CtsD) (Fig 3A). Furthermore, the

proteomics analysis revealed that CtsB, CtsC, CtsD, CtsS, and CtsZ were distributed across all RTM phagosomes, whereas CtsA, CtsH, CtsK, and CtsL showed RTM-specific phagosomal expression (Fig 3B).

Peritoneum-enriched CtsA is a serine-type carboxypeptidase known to be involved in the processing of biologically active peptides including angiotensin I or bradykinin (Timur et al, 2016). This correlates with high efficiency of these cells to generate bradykinin fragment and suggests that CtsA might be the effector responsible for peptide hormone signalling exerted by peritoneal macrophages (Vietinghoff & Paegelow, 2000).

Brain-enriched CtsL has been shown to be involved in microglial processing of invariant chain and remodelling of basal lamina ECM (Gresser et al, 2001; Gu et al, 2015). Interestingly, CtsL also mediates degradation of proteins involved in neurodegeneration such as tau or progranulin (Bednarski & Lynch, 1998; Lee et al, 2017). Our results support the crucial role of CtsL in homeostatic maintenance of neuronal tissue (Xu et al, 2018), and we hypothesize that high phagosomal levels of CtsL in brain-resident macrophages represent specific adaptation to substrates abundant in this tissue environment.

CtsH and CtsK showed lung-specific distribution. Notably, CtsK was the only phagosomal protease whose expression was solely restricted to lung phagosomes (Fig 3A and B). CtsK is one of the most efficient endogenous ECM-degrading enzymes associated mostly with osteoclasts and epithelioid cells participating in remodelling of tissue (Buhling et al, 2001; Dai et al, 2020), whereas CtsH mediates processing of alveolar surfactant proteins B and C (Ueno et al, 2004; Buhling et al, 2011). Collectively, these data demonstrate that phagosomal distribution of cathepsins in distinct RTMs reflects their local substrates and determines their tissue-specific functions.

## Cathepsin K is required for phagosomal proteolytic and collagenolytic activity in lung-resident macrophages

CtsK has been shown to be associated with matrix remodelling in the lungs (Buhling et al, 2002; Knaapi et al, 2015). However, its expression by lung macrophages has so far been controversial (Buhling et al, 2004; van den Brule et al, 2005; Rapa et al, 2006). To further investigate the potential specific role of CtsK in lung-resident macrophages, we measured its enzymatic activity in lung- and peritoneum-resident macrophages and compared it with ubiquitously present CtsD. As a result, CtsD activity was similar in the cell lysate of both RTMs (Fig 3C), whereas CtsK response was specific to lung-resident macrophages (Fig 3D). These data could potentially explain the enhanced phagosomal proteolytic efficiency of lung-resident macrophages compared with peritoneal macrophages (Fig 1E). To address this question, we inhibited CtsK enzymatic activity by a specific CtsK inhibitor L 006235 and measured phagosomal proteolysis in peritoneum- and lung-resident macrophages. The analysis revealed that the enhanced proteolytic efficiency of lung-resident macrophages was indeed dependent on CtsK (Fig 3E). Interestingly, inhibition of CtsK activity also reduced phagosomal BSA proteolysis in peritoneal macrophages. This might be explained that either even small amounts of CtsK (below detection by MS) might enhance BSA proteolytic processing in

peritoneal macrophages or L 006235 inhibitor partially inhibited the activity of other cathepsins during BSA proteolysis in peritoneal macrophages (Fig 3E).

CtsK is one of the most potent collagenases (Dai et al, 2020). Therefore, we next asked whether high activity of CtsK in lung RTMs can contribute to lung tissue remodelling by mediating clearance of phagocytosed collagen, a more specific substrate of CtsK. To this end, we exposed lung- and peritoneum-resident macrophages to collagen-coated beads and measured their phagosomal collagenolytic activity. The analysis revealed that lung-resident macrophages exhibited significantly higher ability to process phagocytosed collagen compared with peritoneal macrophages (Fig 3F). Moreover, the pharmacological inhibition of CtsK enzymatic activity impaired phagosomal collagenolysis, specifically in lung-resident macrophages, whereas collagen processing in peritoneal macrophages stayed intact (Fig 3F). Altogether, CtsK contributes essentially to phagosomal proteolytic and collagenolytic activities, specifically in lung-resident macrophages.

## Lung-resident macrophages possess CtsK-dependent phagosomal machinery tailored for ECM disposal

If CtsK plays a fundamental role in intracellular degradation of collagen in lung-resident macrophages, it is expected to undergo functional changes associated with pathological states such as fibrosis. Lung fibrosis is manifested as an excessive accumulation of connective tissue driven by TGF-$\beta$ (Martinez et al, 2017; Yue et al, 2010). TGF-$\beta$ stimulates production of collagen in lung fibroblasts and inhibits ECM degradation, whereas CtsK has in general a protective function in lung fibroblasts during development of lung fibrosis (Buhling et al, 2004; Srivastava et al, 2008). TGF-$\beta$ has been shown to down-regulate CtsK expression in lung fibroblasts (van den Brule et al, 2005). At the same time, TGF-$\beta$ was reported to be an efficient substrate of CtsK indicating a regulatory role of CtsK towards TGF-$\beta$ signalling (Zhang et al, 2011).

In lung macrophages, we first examined the effect of TGF-$\beta$ on CtsK gene expression and enzymatic activity. As a result, TGF-$\beta$ down-regulated CtsK enzymatic activity in lung-resident macrophages similar to lung fibroblasts. However, unlike lung fibroblasts, it did not have any effect on CtsK gene expression (Figs 4A and S4A). This suggests that CtsK inhibition occurs on post-translational level, possibly via regulation of activating cleavage of procathepsin K in phagosome for which low pH is required (McQueney et al, 1997). To further dissect TGF-$\beta$ impact on CtsK function, we measured phagosomal collagen and BSA protein degradation in lung-resident macrophages stimulated or not either with profibrotic TGF-$\beta$ or with proinflammatory LPS stimuli. The analysis revealed that profibrotic TGF-$\beta$ specifically reduced degradation of collagen (Fig 4B) without affecting phagocytosis (Fig S4B), whereas proinflammatory stimulation decreased proteolytic activity in general (Fig 4B and C). These data indicate that TGF-$\beta$ specifically impacts CtsK-mediated degradation of intracellular collagen in lung-resident macrophages.

Finally, we sought to extend the role of CtsK in phagosomal collagenolysis in human lung–resident macrophages stimulated or not with TGF-$\beta$ and LPS. In line with mouse data, TGF-$\beta$–stimulated lung-resident macrophages exhibited significantly reduced intracellular degradation of collagen in a CtsK-dependent manner

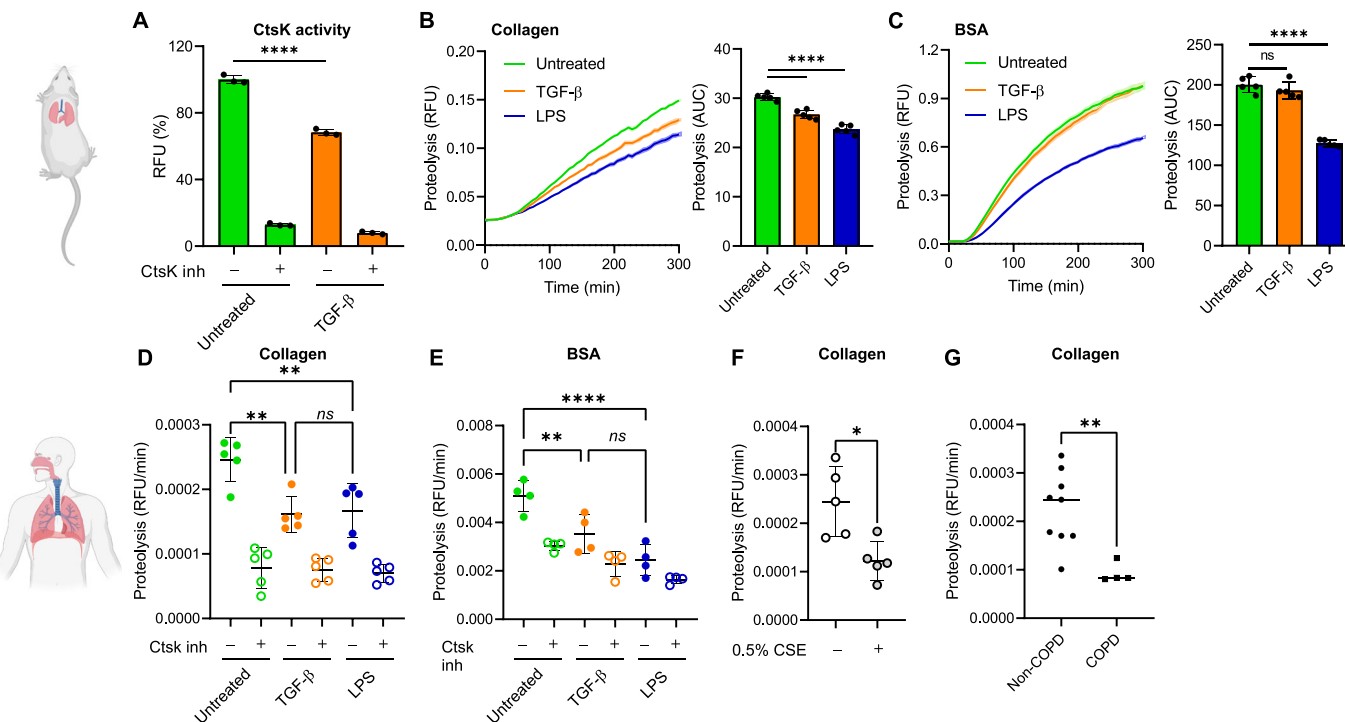

**Figure 4. Lung RTMs possess TGF-β–regulated CtsK-dependent phagosomal machinery tailored for ECM disposal.**
**(A)** Activity of cathepsin K in murine lung RTMs stimulated with TGF-β. Measured from lysates; cells were activated by murine TGF-β (5 ng/ml) 12 h before lysis. CtsK was inhibited by 1 μM L 006235 (CtsK inh) added directly into lysates. **(B, C)** Kinetic measurement of phagosomal degradation of collagen (B) and BSA (C) in murine lung RTMs activated by TGF-β. Cells were activated by murine TGF-β (5 ng/ml) or by *E. coli* LPS (100 ng/ml) 12 h before the exposure to beads. **(D, E)** Phagosomal degradation of collagen (D) and BSA (E) in human lung RTMs activated by TGF-β. Cells were activated by human TGF-β (5 ng/ml) or by *E. coli* LPS (100 ng/ml) 12 h before the exposure to beads. Inhibition of CtsK by 1 μM L 006235 (CtsK inh) was done 1 h before and during proteolysis. Each dot in the graph characterizes a single donor of lung macrophages, n = 4/5. **(F)** Inhibition of phagosomal collagen degradation by cigarette smoke extract in human lung RTMs. Cells were treated with 0.5% cigarette smoke extract 12 h before exposure to beads. Each dot in the graph characterizes a single donor of lung macrophages, n = 5. **(G)** Decreased phagosomal degradation of collagen in human lung RTMs isolated from COPD patients. Each dot in the graph characterizes a single donor of lung macrophages either from healthy donor (n = 9) or from COPD patient (n = 4). Data information: statistical significance of data is denoted on graphs by asterisks where *P < 0.05, **P < 0.01, ****P < 0.0001, or ns, not significant as determined by ANOVA with the post hoc Tukey test (A, B, C, D, E) or by the Mann–Whitney test (F, G). Data showing time-dependent fluorescence (B, C) are normalized to the uptake of collagen/BSA-coated beads and are expressed as means of RFU ± SEM or AUC ± SD. Data in (A) are shown as means of normalized fluorescence ± SD. Data in (D, E, F, G) are normalized to the uptake of collagen/BSA-coated beads and are expressed as the slope of proteolytic signal during the linear phase characterized by relative fluorescence intensity (RFU) values versus time (min) (RFU/min) ± SD. Data in (A, B, C) are representative of three replicates. Illustrations were created with BioRender.com.

(Fig 4D and E). These data collectively indicate that TGF-β regulates CtsK-mediated phagosomal collagen degradation independently from classical endocytic–proteolytic pathways.

### Cathepsin K–dependent collagenolytic inactivity in lung RTMs is associated with COPD pathology

The enhanced collagen clearance mediated by lung macrophages would be of critical importance during exaggerated remodelling of lung tissue, such as in COPD (Guieu & Hellon, 1980). One of the pathological hallmarks of COPD is emphysema. It is characterized by destruction of alveolar walls mediated by elastases secreted by immune cells (Abboud & Vimalanathan, 2008). This generates collagen fragments, which are proinflammatory and further stimulate tissue damage (Weathington et al, 2006). We hypothesized that inefficient clearance of partially digested collagen by lung-resident macrophages might be another factor involved in COPD pathology. Cigarette smoke is one of the major causes associated with the development of COPD (Barnes et al, 2015). We therefore

tested whether cigarette smoke extract (CSE) affects phagosomal degradation of collagen in human lung–resident macrophages. CSE significantly decreased intracellular processing of collagen-coated beads, even in concentrations that do not impact phagocytic activity (Figs 4F and S4C). Next, we determined collagenolytic properties of lung-resident macrophages isolated from COPD patients (Table S2). The analysis revealed that COPD-derived lung-resident macrophages exhibited reduced CtsK-dependent degradation of phagocytosed collagen compared with non-COPD control (Fig 4G). This suggests that lung RTMs might contribute to COPD pathology by slower intracellular removal of proinflammatory ECM fragments.

Taken together, we show here that distribution of cathepsins in RTM phagosomes is adapted to abundant substrates of residing tissues and that lung RTMs use a unique CtsK-dependent phagosomal pathway for clearance of phagocytosed collagen. Given that TGF-β, CSE, and COPD pathology all reduce CtsK-mediated collagenolysis in lung RTMs, we assume that lung tissue insults decrease the ability of lung macrophages to dispose ECM fragments and that their accumulation may exaggerate chronic lung pathologies.

**Life Science Alliance**

# Materials and Methods

### Isolation of murine RTMs

All RTMs were isolated from 6- to 8-wk-old C57BL/6 mice. All mice were maintained under specific pathogen-free conditions, and experiments were approved and carried out according to the guidelines set out by the Regional Animal Ethics Committee with approval no. 2947/20. Peritoneal RTMs were harvested by injecting PBS into the peritoneal cavity. Lung RTMs were obtained by homogenizing lungs using Lung Dissociation Kit (Miltenyi Biotec) followed by erythrocyte lysis. Cell suspension was then seeded into a bacterial plastic dish, and lung RTMs were obtained as an adherent cell fraction after overnight cultivation. Brain RTMs were isolated by homogenizing brains using Adult Brain Dissociation Kit (Miltenyi Biotec) according to the manufacturer's instructions. Obtained cells were enriched for CD11b[+] cells using CD11b MicroBeads (Miltenyi Biotec) and seeded into poly-lysine–coated plastic. After isolation, all RTMs were cultivated in DMEM/F-12 with 10% FBS, 100 U/ml penicillin, and 100 $\mu$g/ml streptomycin (all from Gibco), and LCCM for 3 d, which was washed away the day before the experiment.

### Isolation of human lung macrophages

Human lung RTMs were isolated as previously described (Andelid et al, 2021). In brief, they were harvested by injecting excess of PBS into human lung tissue until cell suspension started to seep out. This was repeated several times until PBS coming out from the tissue appeared completely clear. Combined cell suspension was centrifuged, and erythrocytes were lysed by ACK lysis buffer (Gibco). Human RTMs were resuspended in XVivo10 medium (Lonza) supplemented with 2 mM glutamine and penicillin/streptomycin (10,000 U/ml; Gibco) seeded as required, and used in experiments the next day. Human lung samples were obtained from patients with or without COPD undergoing lung cancer resection surgery at the Department of Cardiothoracic Surgery at Sahlgrenska University Hospital. All human lung tissue samples were acquired in accordance with hospital and AstraZeneca ethical guidelines with written consent from all patients. The ethics committee approval number was 1026-15.

### Phagocytosis assay

RTMs were seeded into a 384-well plate at a density of $2.5 \times 10^4$ cells per well 24 h before the experiment. Cells were treated with 3.0-$\mu$m silica beads (Kisker Biotech) coated by AF488-conjugated BSA diluted at a ratio 1:100 by complete DMEM/F-12 medium and kept for 30 min at 37°C. Treatment was stopped by aspirating medium followed by the addition of trypan blue to quench the fluorescence of extracellular beads. After aspirating trypan blue, fluorescence was measured on a SpectraMax i3x plate reader (Molecular Devices) using excitation/emission wavelengths of 488/525 nm. When indicated, RTMs were treated with cytochalasin D (10 $\mu$M; Sigma-Aldrich) for 1 h before and during the treatment.

### Phagosome functional assays

For phagosomal proteolysis assay, RTMs were seeded into a 384-well plate and treated with 3.0-$\mu$m silica beads (Kisker Biotech) coated either by DQ Green BSA or by DQ Collagen, type I (both from Thermo Fisher Scientific). Beads were diluted at a ratio 1:100 by binding buffer (1 mM CaCl$_2$, 2.7 mM KCl, 0.5 mM MgCl$_2$, 5 mM dextrose, 10 mM Hepes, and 5% FBS in PBS, pH 7.2) and kept with cells for 5 min. After treatment, beads were replaced by warm binding buffer and real-time fluorescence was measured at 37°C on a SpectraMax i3x plate reader using excitation/emission wavelengths of 470/525 nm. The experimental set-up for phagosomal acidification assay was similar except that beads coated by BSA conjugated with pHrodo Red AM (Thermo Fisher Scientific) were used and the measured fluorescence was at 560/590 nm. All beads were labelled by AF647, and the reported fluorescence was corrected for bead loading by normalizing to AF647 signal (ex/em 640/665 nm). Quantitative data are presented either as area under the curve or as the slope of proteolytic signal during the linear phase characterized by relative fluorescence intensity (RFUs) values versus time (min) (RFU/min). When indicated, RTMs were treated with bafilomycin A1 (100 nM; Enzo) or L 006235 (1 $\mu$M; Tocris) for 1 h before and during the treatment and with murine TGF-$\beta$ (5 ng/ml; R&D) or *E. coli* 055:B5 LPS (100 ng/ml; InvivoGen) 12 h before treatment.

### Uptake of apoptotic and necrotic cells

Mouse embryonic fibroblasts were treated with 50 $\mu$M cycloheximide for 12 h to induce apoptosis or subjected to four freeze/thaw cycles to induce necrosis as previously described (Martinez et al, 2011). Apoptotic and necrotic cells were then washed with ice-cold PBS and labelled by 5 $\mu$M SYTO17 Red Fluorescent nucleic acid dye (Thermo Fisher Scientific) for 30 min at 4°C in dark. After staining, cells were washed with ice-cold PBS, resuspended in complete DMEM/F-12 medium, and added to RTMs seeded into a 384-well plate at an approximate ratio 10:1 (cells: RTM). After 4 h at 37°C, RTMs were washed extensively with PBS and the fluorescence signal of SYTO17 Red was measured on a SpectraMax i3x plate reader. As a negative control, to distinguish internalized apoptotic/necrotic cells from extracellular attached to macrophage surface, we inhibited phagocytosis by treating cells at a four-degree condition (Fig S4D).

### Cathepsin activity assays

Cathepsin activity was determined by Cathepsin K and Cathepsin D Activity Assay Kits (both from Abcam) according to the manufacturer's instructions. Briefly, $1 \times 10^6$ RTMs were washed with PBS, lysed on ice for 10 min using supplied lysis buffer, and centrifuged (13,000*g*, 5 min, 4°C). Supernatants were mixed with cathepsin-specific fluorescent substrate and optionally with 1 $\mu$M L 006235 or 1 $\mu$M pepstatin A (Tocris) and incubated for 1 h at 37°C. Fluorescence was measured at excitation/emission wavelengths of 400/505 nm and 328/460 nm for CtsK and CtsD substrates, respectively. When indicated, RTMs were treated with murine TGF-$\beta$ (5 ng/ml; R&D) 12 h before lysis—TGF-$\beta$ effect was controlled by down-regulation of CD80 (Fig S4E) (Zhang et al, 2016).

## CSE preparation and cell treatment

Smoke from two 2R4F reference cigarettes (University of Kentucky) was passed through 25 ml of serum-free PBS for a total time of 10 min at a flow rate of 0.07 litres/min. Obtained 100% CSE was filtered through 0.22-$\mu$m filters and used directly or snap-frozen, and stored at −80°C. For CSE treatment, cell cultivation medium was replaced by medium-diluted CSE as indicated and kept with cells for 12 h before further experiments.

## Flow cytometry

RTMs were collected at $5 \times 10^5$ per FACS tube, washed with FACS buffer (1% BSA/5 mM EDTA/PBS, pH 7.2), and incubated in blocking buffer (1:100 anti-CD16/CD32 in FACS buffer) for 15 min at 4°C. After blocking, RTMs were washed with FACS buffer and stained by anti-CD45-BV711 (BD), F4/80-PE (BioLegend), CD11b-FITC (BD), Siglec-F-PE (BD), CD11c-APC-Cy7 (BioLegend), or CD80-APC (BD) at a dilution of 1:200 by FACS buffer for 30 min at 4°C. After washing with FACS buffer, cells were analysed by LSRFortessa X-20 (BD) and FlowJo software, v 10.8.1.

## Isolation of phagosomes

RTMs isolated from 5–10 mice were washed three times with PBS and then incubated for 30 min at 37°C with 1.0 $\mu$m carboxylated magnetic beads (Dynabeads; Thermo Fisher Scientific) diluted at 1:20 ratio by complete DMEM/F-12 medium. Cells were then washed with ice-cold PBS and scraped, and cell pellets were washed two times with ice-cold PBS and once with hypotonic lysis buffer (250 mM sucrose/3 mM imidazole, pH 7.4). Cells were lysed in hypotonic lysis buffer containing inhibitors of phosphatases and proteases (EDTA-free; Thermo Fisher Scientific) and 250 U/ml of nuclease (Thermo Fisher Scientific) by 15 strokes in a Dounce homogenizer. Tubes with lysates were then transferred into a magnetic holder, and supernatants were removed. Phagosomes were washed four times with ice-cold PBS and centrifuged (13,000$g$, 5 min, 4°C) after the final wash. The supernatant was discarded, and pelleted phagosomes were stored at −80°C.

## Phagosome sample preparation for proteomics

Phagosome pellets were dissolved in 25 $\mu$l of 5% SDS/50 mM triethylammonium bicarbonate (TEAB) buffer (pH 7.6) and centrifuged. Supernatants were transferred into new tubes, and protein concentrations were determined by BCA assay (Sigma-Aldrich). ~6 $\mu$g of phagosomal proteins from each sample was reduced by 10 mM tris(2-carboxyethyl)phosphine for 30 min at RT and alkylated by 10 mM iodoacetamide for 30 min in dark. The excess of iodoacetamide was quenched by the addition of tris(2-carboxyethyl)phosphine (final concentration 20 mM) for 15 min. Protein digestion was done using S-Trap (ProtiFi) according to the manufacturer's instructions. Briefly, samples were acidified by phosphoric acid, mixed with 6 vol of 90% methanol (MeOH)/100 mM TEAB, and loaded onto the S-Trap phase. Columns were then washed four times with 90% MeOH/100 mM TEAB, and captured proteins were digested by 1 $\mu$g of trypsin (Pierce) in 50 mM TEAB at

37°C overnight. After digestion, peptides were eluted by consecutive washes of 50 mM TEAB, 0.2% formic acid (FA), and 50% acetonitrile/0.2% FA and the combined eluate was evaporated to ~20 $\mu$l. Samples were acidified by TFA, desalted using reversed-phase spin columns (Thermo Fisher Scientific), and dried on Speed-Vac.

## LC-MS/MS analysis

An Easy nanoLC 1200 liquid chromatography system connected to an Orbitrap Lumos Tribrid mass spectrometer (Thermo Fisher Scientific) was used for proteomics analyses. Peptides were first introduced onto a trap column (PepMap 100 C18, 5 $\mu$m, 0.1 × 20 mm) and then separated on an in-house packed analytical column (ReproSil-Pur C18, 3 $\mu$m, 0.075 × 330 mm, Dr Maisch) using gradient (0.2% FA in water as phase A; 0.2% FA in acetonitrile as phase B) running from 6% to 35% B in 167 min and from 35% to 100% B in 3 min, at a flow rate of 300 nl/min. Positive ion MS scans were acquired at a resolution of 120,000 within $m/z$ range of 400–1,600 using AGC target $5 \times 10^5$ and maximum injection time 50 ms. MS/MS analysis was performed in data-dependent mode with a top speed cycle of 1 s for the most intense multiply charged precursor ions. MS precursors above 50,000 threshold were isolated by quadrupole using 0.7 $m/z$ isolation window and then fragmented in ion trap by collision-induced dissociation at a collision energy of 35%. Dynamic exclusion was set to 60 s with 10 ppm tolerance. MS/MS spectra were acquired by ion trap using AGC target $5 \times 10^5$ and maximum injection time 35 ms.

## Proteomics data search

Proteomics datasets were processed by MaxQuant, ver. 1.6.10.43 (Cox & Mann, 2008). Data were searched against reference proteome of *Mus musculus* (UP000000589; 7 August 2020) downloaded from UniProt. The MaxQuant-implemented database was used for the identification of contaminants. Protein identification was done using these MaxQuant parameters as follows: mass tolerance for the first search 20 ppm, for the second search from recalibrated spectra 4.5 ppm; maximum of two missed cleavages; maximal charge per peptide $z = 7$; minimal length of peptide 7, maximal mass of peptide 4,600 D; and carbamidomethylation (C) as fixed and acetylation (protein N-term) and oxidation (M) as variable modifications with the maximum number of variable modifications per peptide set to 5. Trypsin with no cleavage restriction was set as a protease. Mass tolerance for fragments in MS/MS was 0.5 D, taking the eight most abundant peaks per 100 D for search (with enabled possibility of cofragmented peptide identification). FDR filtering on peptide spectrum match was 0.01, and only proteins with at least one identified unique peptide were considered further. Proteins were quantified using the MaxLFQ function (Cox et al, 2014) with at least one peptide ratio required for pairwise comparisons of protein abundance between samples. Proteins identified as contaminants were removed before any further interpretation of data.

## Data interpretation

Annotation of proteins by gene ontology was done in Perseus, ver. 1.6.2.1 (Tyanova et al, 2016), and the enrichment was determined by the Fisher exact test. For protein clustering, LFQ intensities of the

given protein in each sample were divided by the median protein intensity and ratios were log-transformed. Values were clustered by K-means in R, and the data were visualized by the "pheatmap" package employing hierarchical clustering of samples using the Euclidean distance. Only proteins identified in all three replicates in each tissue were considered for the clustering. The triwise plot and rose plots were constructed using the "triwise" package (van de Laar et al, 2016) in R by taking log-transformed protein LFQ intensities averaged from at least two replicates per tissue.

### Western blot

Phagosome samples were lysed in 5% SDS/50 mM TEAB, and protein concentration was determined by BCA assay. For electrophoresis, samples were mixed with 4× Laemmli sample buffer and $\beta$-mercaptoethanol, heated to 95°C for 5 min, and run on a NuPAGE 4–12% Bis-Tris gel (Life Technologies). Proteins were transferred onto the PVDF membrane using the Mini Trans-Blot Cell system (Bio-Rad). Membranes were blocked by 5% non-fat dried milk in Tris-buffered saline/0.1% Tween-20 (TBS-T) for 1 h at RT, and incubated with primary antibodies in 3% BSA in TBS-T at 4°C overnight and with secondary antibodies in 5% milk/TBST for 1 h at RT. After incubation with HRP-labelled secondary antibodies, proteins were detected using ECL and X-ray films. The following antibodies were used: anti-LAMP1 (Cell Signaling Technology), tubulin (Abcam), and Rab7 (Cell Signaling Technology).

### qRT–PCR analysis and CtsK expression

Lung RTMs were stimulated with TGF-$\beta$ (5 ng/ml) for 12 h before lysis and isolation of RNA using RNeasy Mini Kit (QIAGEN) according to the manufacturer's instructions. RNA was reverse-transcribed by High-Capacity cDNA Reverse Transcription Kit (Thermo Fisher Scientific), and qRT-PCR analysis was done on LightCycler 96 (Roche) using TaqMan Universal Master Mix II (Thermo Fisher Scientific). Murine *Ctsk* was detected by TaqMan probe (Mm00484039_m1; Thermo Fisher Scientific), and its expression was normalized to *Actb* (Mm02619580_g1; Thermo Fisher Scientific) and to the levels in untreated cells ($\Delta\Delta$Ct method).

### Statistical analysis

Statistical analysis was performed in GraphPad Prism software 8.3.1.

## Data Availability

The mass spectrometry proteomics data have been deposited to the ProteomeXchange Consortium (http://proteomecentral.proteomexchange.org) via the PRIDE partner repository (Perez-Riverol et al, 2019) with the dataset identifier PXD033010.

## Supplementary Information

## Acknowledgements

This work was funded by the Knut and Alice Wallenberg Foundation and the Wallenberg Centre for Molecular and Translational Medicine (dedicated to A Härtlova), University of Gothenburg, Sweden, Cancerfonden 19 0352 (dedicated to A Härtlova), Wilhelm och Martina Lundgrens Stiftelser 2020, 2021 (dedicated to A Härtlova), and MH CZ – DRO (UHHK, 00179906, dedicated to I Fabrik). We would like to thank the Proteomics Core Facility at Sahlgrenska Academy at the University of Gothenburg, Sweden.

### Author Contributions

I Fabrik: formal analysis, supervision, methodology, and writing—original draft, review, and editing.
O Bilkei-Gorzo: data curation and methodology.
M Öberg: data curation and methodology.
D Fabrikova: data curation and methodology.
J Fuchs: data curation and methodology.
C Sihlbom: data curation and methodology.
M Göransson: data curation, supervision, methodology, and project administration.
A Härtlova: conceptualization, resources, supervision, funding acquisition, project administration, and writing—original draft, review, and editing.

### Conflict of Interest Statement

The authors declare that they have no conflict of interest.

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
