## [Reviewer comments · Life Science Alliance]

Life Science Alliance

Lung macrophages utilize cathepsinK-dependent phagosomal machinery to degrade intracellular collagen

Ivo Fabrik, Orsolya Bilkei-Gorzo, Maria Öberg, Daniela Fabrikova, Johannes Fuchs, Carina Sihlbom, Melker Göransson, and Anetta Härtlova

DOI: <https://doi.org/10.26508/lsa.202201535>

Corresponding author(s): Anetta Härtlova, University of Gothenburg and Anetta Härtlova, University of Gothenburg

Review Timeline:

Submission Date:	2022-05-26
Editorial Decision:	2022-07-05
Revision Received:	2022-11-23
Editorial Decision:	2022-12-15
Revision Received:	2022-12-22
Accepted:	2023-01-03

Transaction Report:

July 5, 2022

Re: Life Science Alliance manuscript #LSA-2022-01535-T

Dr. Anetta Härtlova
University of Gothenburg, Institute of Biomedicine, Wallenberg Centre for Molecular and Translational Medicine (WCMTM)
Microbiology and Immunology
Medicinaregatan 7A
Gothenburg, Sweden/Västra Gotaland 40530
Sweden

Dear Dr. Härtlova,

Thank you for submitting your manuscript entitled "Lung macrophages utilize cathepsin K-dependent phagosomal machinery to degrade intracellular collagen" to Life Science Alliance. The manuscript was assessed by expert reviewers, whose comments are appended to this letter. We invite you to submit a revised manuscript addressing the Reviewer comments.

Thank you for this interesting contribution to Life Science Alliance. We are looking forward to receiving your revised manuscript.

Sincerely,

B. MANUSCRIPT ORGANIZATION AND FORMATTING:

Reviewer #1 (Comments to the Authors (Required)):

The manuscript „Lung macrophages utilize unique cathepsin 1 K-dependent phagosomal 2 machinery to degrade intracellular collagen" by Fabrik and colleagues aims to compare phagocytic capabilities of macrophages of different origins. They found that cathepsin K is the main peptidase in the intracellular destruction of collagen and is mainly active in lung macrophages. The manuscript is interesting and well written and I was rather enthusiastic about this manuscript; however, I have several issues with this manuscript especially with methodological inconsistencies.

- The macrophages harvested from the mice were cultured in L929 supernatants for "3-5 days". Why? L929 release M-CSF (CSF1), i.e all macrophages are CSF1-stimulated and thus M2 differentiated. As seemingly the incubation time was not consistently used (3-5 days....), the macrophages might be used at different differentiation steps. This should be mentioned and discussed.
- Was this culture protocol used in all other RTM experiments? This is also not mentioned.
- In addition, the human macrophages were seemingly used without L929 supernatant culture, thus they were used under different conditions.
- The strategy to identify the macrophages from the lung is different from the strategy in the other organs. Why? Again, this is not mentioned nor discussed. In addition, it is not clear, which subpopulation is analysed: SiglecF-pos or neg or both?
- I do not understand, how you can get a value of $p < 0.001$ with 2 or 3 replicates. With this low replicate numbers ANOVA is not reliable.
- It is a nice idea to block Fc-receptors with antibodies, however as long as it is not clear whether or not these antibodies are functional and target the Ig-binding motive of the Fc-receptors, these antibodies are useless.
- The LSRFortessa X-20 is the machine but the software used for analysis should also be mentioned.

Reviewer #2 (Comments to the Authors (Required)):

The manuscript by Fabrik I et al., utilizes a quantitative proteomic approach to study phagosome maturation in resident tissue macrophages (RTMs). The authors provide data suggesting that cathepsin K (CtsK) is required for collagen degradation and that this process is regulated by TGF-beta. Finally, they provide evidence that phagosomal activity is reduced in patients with COPD and in macrophages stimulated with smoke extract. While aspects of the work are interesting, it is well written, discusses the previous literature and the figures are nicely presented, the work contains several technical and methodological limitations/issues that weaken the authors conclusions. These unfortunately, ultimately diminish the significance of the claims in its current format.

Major comments.

1) In the methods, the authors state that all RTMs were cultured with L929 media for 3-5 days. I find this very surprising. Aside from containing M-CSF, proteomic characterization of L929 media indicates it contains various cytokines and chemokines (PMID: 33853969). Given that the authors are examining RTMs, this begs the question as to how much the data reflects RTM phenotypes in vivo. In my opinion it would have made much more sense to simply culture the cells for 24h in RPMI/FCS prior to experiments.

2) Fig. EV1: No viability stain has been included. Furthermore, resident brain macrophages (microglia) are routinely gated as CD45^{int} CD11b⁺ cells and the gating strategy for lung macrophages suggests these cells represent a mixture of alveolar macrophages, interstitial macrophages, monocytes and DCs (PMID: 23672262) activated by the L929 (please see point 1). Surprisingly, the % of SiglecF⁺, CD11c⁺ cells is low and the majority of cells are SiglecF⁻ but CD11c⁺, possibly indicating DC contamination.

3) The authors should present Annexin V/7AAD data or equivalent experimental evidence demonstrating the degree of apoptosis and necrosis in the mouse embryonic fibroblasts used in Fig. 1A and B. Furthermore, in this data (Fig. 1A/B) no 4 deg C control or cytochalasin D has been included (and is not listed in the methods), therefore how can the authors distinguish between binding and uptake? Additionally, I don't understand why SYTO dye was used for these assays (which binds nucleic

acids). It would make more sense to use PH Rodo as this would show direct uptake.

4)The authors write, "Inhibition of CtsK activity also slightly reduced phagosomal proteolysis of peritoneal macrophages... (lines 180-181)." In my opinion, this looks like a significant inhibition (Fig. 3E). The test used was an ANOVA with post-hoc Tukey. Could these authors please add the significance/p value. Given Ctsk was not detected by MS in peritoneal macrophages, this suggests that the Ctsk inhibitor may have non-specific effects. Thus, it would important to back up the inhibitor data with experiments utilizing Ctsk deficient macrophages (PMID: 15161653).

5) Given that TGF-beta lowers Ctsk expression in lung fibroblasts (PMID: 16045809), could it have a dual effect on both enzymatic activity and expression in lung macrophages? Do the authors have experimental evidence that these effects are specific for activity?

6) The relevance of CD80 expression decreases by TGF-beta in Fig EV3A is not clear in the text.

7)The methods for isolation of human lung macrophages is not clear. Was the tissue digested? Additionally, no quality control or FACS plots are shown to indicate viability and phenotype of the human cells isolated (PMID: 28769058).

8) It is not clear if the data from Fig. 4F represents technical replicates or individual donors. Furthermore, it would be important to present the data as RFU in the supplement (similar to Fig. 4B) so the reader can determine the magnitude of delta RFU and easily compare the data across the Figure. Similar point for the data shown in Fig. 4D/E.

9) Only 2 patients are shown in Fig. 4G. To strengthen the claim that COPD macrophages exhibit reduced Ctsk dependent collagen degradation versus controls, it would be important to significantly enhance this patient cohort and additionally provide demographics (e.g. sex, age etc between controls and COPD patients).

10)Tumor associated macrophages have been shown to degrade collagen via uptake through the mannose receptor (PMID: 29281816). The authors should cite this paper. Could they speculate on whether the mannose receptor might play a similar role in lung macrophages? Additionally, it would interesting to examine if other M2 stimuli aside from TGF-beta (e.g. IL-4 or IL-10) regulate Ctsk activity as this may strengthen the paper.

Minor points

1) The authors mention that CtsD activity is similar between the cell lysates of both RTMs (Fig. 3C) (lines 174-175). However, pepstatin A is much more effective at inhibiting CtsD in peritoneal than lung macrophages. Can the authors comment on this? Might this reflect expression levels of the different cathepsins (in this case CtsD) between the different RTMs?

2) The authors should discuss the significance of their findings in relation to literature suggesting that during lung fibrosis, fibroblasts and not lung macrophages are the main cells upregulating CtsK and taking up collagen (PMID: 15161653) and that CtsK has been shown to regulate TGF-beta (PMID: 21627832).

3) The beta is missing from Fig. 4A

Reviewer 1:

The manuscript „Lung macrophages utilize unique cathepsin 1 K-dependent phagosomal machinery to degrade intracellular collagen" by Fabrik and colleagues aims to compare phagocytic capabilities of macrophages of different origins. They found that cathepsin K is the main peptidase in the intracellular destruction of collagen and is mainly active in lung macrophages. The manuscript is interesting and well written and I was rather enthusiastic about this manuscript; however, I have several issues with this manuscript especially with methodological inconsistencies.

We thank the reviewer for encouraging comments. We agree that there were misunderstandings about using L929 and methodological inconsistencies regarding characterization lung macrophages. Therefore, we have now clarified the culture condition with L929, have added experiments of phagosomal acidification and proteolysis in RTMs cultured without L929 and F4/80⁺ and CD11b⁺ staining of lung macrophages.

Comment#1: The macrophages harvested from the mice were cultured in L929 supernatants for "3-5 days". Why? L929 release M-CSF (CSF1), i.e all macrophages are CSF1-stimulated and thus M2 differentiated. As seemingly the incubation time was not consistently used (3-5 days....), the macrophages might be used at different differentiation steps. This should be mentioned and discussed.

Response 1: We thank the reviewer for an input. We apologize, there was a misunderstanding in the methods section. We developed phagocytic assays of different RTMs with or without L929. We have now added our data of phagocytic characterization of distinct RTMs without L929 into the supplementary figure (**Supplementary figure EV2**). There were no phenotypic differences between different RTMs with or without L929. However, RTMs showed better adherence and rate of phagocytosis in the presence of L929. We are aware of limitations of our study. Due to demands of proteomics approach, we decided to cultivate all mouse RTMs in the presence for L929 for 3 days, L929 was washed away the day before induction of phagocytosis. Moreover, we further confirmed main results of mouse lung macrophages in the human system without presence of L929. We amended the text.

Comment#2: Was this culture protocol used in all other RTM experiments? This is also not mentioned.

Response 2: The same cultivation protocol was used for all murine RTMs. We amended the text now.

Comment#3: In addition, the human macrophages were seemingly used without L929 supernatant culture, thus they were used under different conditions

Response 3: Human lung macrophages were used without L929. L929, which produce murine growth factors, was used to compare all different mouse RTMs as explained in the Response 1.

Comment#4: The strategy to identify the macrophages from the lung is different from the strategy in the other organs. Why? Again, this is not mentioned nor discussed. In addition, it is not clear, which subpopulation is analysed: SiglecF-pos or neg or both?

Response 4: We thank the reviewer for the comment. Lung macrophages were enriched by digestion of lung tissue using Miltenyi Lung Dissociation kit followed by adherence. They are combination of interstitial macrophages characterized by F4/80⁺, Cd11b⁺ and intermediate CD11c⁺, Siglec F negative and alveolar macrophages characterized by F4/80⁺, Cd11b⁺ low, CD11c⁺, Siglec F positive (<https://pubmed.ncbi.nlm.nih.gov/30867240/>). Our lung macrophages are mostly composed of interstitial lung macrophages (CD11c⁺, Siglec F negative) with smaller fraction of alveolar macrophages (CD11c⁺, Siglec F positive). We have added F4/80⁺ and CD11b⁺ to be consistent with the other RTMs to the **Supplementary Figure EV1**. The mixture of both lung macrophage subpopulations was used in all experiments with murine RTMs and we amended the text. In addition to FACS analysis, MS/MS analysis confirmed high enrichment of CD11c (Itgax) while CD11b (Itgam) is less present in the lung phagosomes compared to peritoneal and brain phagosomes. Siglec F is specifically enriched in the lung phagosome (*Figure 1 below in Rebuttal*), which corresponds to cell surface expression.

Figure 1: Heatmap of log₂ phagosomal enrichment of cell surface markers.

Comment#5: I do not understand, how you can get a value of $p < 0.001$ with 2 or 3 replicates. With this low replicate numbers ANOVA is not reliable.

Response 5: As previously published (<https://doi.org/10.15252/emj.2021108970>), (PMID: 31028084), intraphagosomal proteolysis and acidification assays and CtsD/CtsK assays are plotted as representative graphs of two or three independent biological experiments with 3-5 technical replicates. Due to low yield of brain macrophages, one biological replicate was RTMs isolated from 3 mice. To ensure applicability of ANOVA, we performed normality test (Shapiro-Wilk) and test of equal variance/SD (Brown-Forsythe), we amended the text in material and methods section.

Comment#6: It is a nice idea to block Fc-receptors with antibodies, however as long as it is not clear whether or not these antibodies are functional and target the Ig-binding motive of the Fc-receptors, these antibodies are useless.

Response 6: We thank the reviewer for the comment. We demonstrated the efficiency of Fc blocking by FCRa (CD64) staining with and without blocking agent (Figure 2 in Rebuttal).

Figure 2: The effect of Fc-blocking on Fc-Ra (CD64) staining plotted as a mean fluorescence intensity (MFI).

Comment#7: The LSRFortessa X-20 is the machine but the software used for analysis should also be mentioned.

Response 7: We thank the reviewer for the comment. We added FlowJo software 10.8.1. We amended the text.

Reviewer 2:

The manuscript by Fabrik I et al., utilizes a quantitative proteomic approach to study phagosome maturation in resident tissue macrophages (RTMs). The authors provide data suggesting that cathepsin K (CtsK) is required for collagen degradation and that this process is regulated by TGF-beta. Finally, they provide evidence that phagosomal activity is reduced in patients with COPD and in macrophages stimulated with smoke extract. While aspects of the work are interesting, it is well written, discusses the previous literature and the figures are nicely presented, the work contains several technical and methodological limitations/issues that weaken the authors conclusions. These unfortunately, ultimately diminish the significance of the claims in its current format.

We thank the reviewer for supporting comments and we agree that there were misunderstandings about using L929 and methodological limitations. Therefore, we have clarified the culture condition with L929, we have performed additional experiments to address methodological issues and we have added more data of human lung macrophages derived from healthy donors stimulated or not with cytokines or cigarette smoke and COPD patients.

Comment#1: 1) In the methods, the authors state that all RTMs were cultured with L929 media for 3-5 days. I find this very surprising. Aside from containing M-CSF, proteomic characterization of L929 media indicates it contains various cytokines and chemokines (PMID: 33853969). Given that the authors are examining RTMs, this begs the question as to how much the data reflects RTM phenotypes in vivo. In my opinion it would have made much more sense to simply culture the cells for 24h in RPMI/FCS prior to experiments.

Response 1: We thank the reviewer for an input. We apologize, there was a misunderstanding in the methods section. We developed phagocytic assay of different RTMs with or without L929. We have now added our data of phagocytic/phagosomal characterization of distinct RTMs without L929 into the supplementary figure (**Supplementary figure EV2**). There were no phenotypic differences between different RTMs with or without L929. However, RTMs showed better adherence and rate of phagocytosis in the presence of L929. We are aware of limitations of our study. Due to demands of proteomics approach, we decided to cultivate all mouse RTMs in the presence for L929 for 3 days, L929 was washed away the day before induction of phagocytosis. Moreover, we further confirm main results of mouse lung macrophages in the human system without presence of L929. We amended the methods and main text.

Comment#2: Fig. EV1: No viability stain has been included. Furthermore, resident brain macrophages (microglia) are routinely gated as CD45int CD11b+ cells and the gating strategy for lung macrophages suggests these cells represent a mixture of alveolar macrophages, interstitial macrophages, monocytes and DCs (PMID: 23672262) activated by the L929 (please see point 1). Surprisingly, the % of SiglecF+, CD11c+ cells is low and the majority of cells are SiglecF- but CD11c+, possibly indicating DC contamination.

Response 2: The viability of RTMs was validated by trypan blue before all functional assays. In overall, only cells with over 95% of viability were used for the experiments. FACS analysis was done on fixed RTMs. Regarding brain macrophages, the FACS analysis was performed on already enriched CD11b+ macrophages using magnetic-activated cell sorting (MACS) which were positive for CD45 (**Supplementary Figure EV1**). Classical gating of brain macrophages, microglia (CD45int CD11b+ cells) is usually done from the whole brain. Lung macrophages were enriched by digestion of lung tissue using Miltenyi Lung Dissociation kit followed by adherence to plastic overnight. They are a combination of interstitial macrophages characterized by F4/80+, Cd11b+ and CD11c+, Siglec F negative and alveolar macrophages characterized by F4/80+, Cd11b+ low, CD11c+, Siglec F positive (<https://pubmed.ncbi.nlm.nih.gov/30867240/>). Our lung macrophages are mostly composed of interstitial lung macrophages (CD11c+, Siglec F negative) with smaller fraction of alveolar macrophages (CD11c+, Siglec F positive). We added now F4/80+ and CD11b+ to be consistent with the other RTMs. These data exclude dendritic cells and monocytes since we only utilized adherent cells. In addition, MS/MS analysis confirmed the expression of F4/80 and CD64 with comparable abundance to peritoneal macrophages (*Figure 1 below in Rebuttal*) confirming macrophage phenotype of lung macrophages. Moreover, as described by Drainer et al. (<https://pubmed.ncbi.nlm.nih.gov/30867240/>) SiglecF negative interstitial lung macrophages obtained by lung tissue digestion and overnight adhesion are irresponsive to M-CSF which correlates with our data showing that phagocytic characteristics of lung RTMs with and without L929 were similar.

Figure 1: Heatmap of log₂ phagosomal enrichment of cell surface markers.

Comment#3) The authors should present Annexin V/7AAD data or equivalent experimental evidence demonstrating the degree of apoptosis and necrosis in the mouse embryonic fibroblasts used in Fig. 1A and B. Furthermore, in this data (Fig. 1A/B) no 4 deg C control or cytochalasin D has been included (and is not listed in the methods), therefore how can the authors distinguish between binding and uptake? Additionally, I don't understand why SYTO dye was used for these assays (which binds nucleic acids). It would make more sense to use PH Rodo as this would show direct uptake.

Response 3: We thank the reviewer for the input. We utilized well-established phagocytic assay of apoptotic/necrotic cells previously described (PMID: 31028084). As Reviewer suggested, we have now demonstrated the degree of apoptosis and necrosis in mouse MEFs by Annexin V/Propidium iodide (PI) staining (*Figure 2 in Rebuttal*). Necrotic cells were positive for PI and Annexin V staining while apoptotic cells were positive for Annexin V and significantly less for PI. To further distinguish internalized apoptotic/necrotic cells from extracellular attached to macrophage surface, we inhibited phagocytosis by keeping cells at four degree as suggested by the Reviewer2. As shown in **Supplementary Figure EV4E**, the assay reliably quantifies internalized apoptotic/necrotic cells. SYTO dye was used to label necrotic and apoptotic cells as previously described by Douglas Green (<https://www.ncbi.nlm.nih.gov/pmc/articles/PMC3198353/>). pH RODO is a pH-sensitive dye for determination of phagosomal acidification (<https://pubmed.ncbi.nlm.nih.gov/31028084/>). Therefore, pH Rodo is not suitable dye for determination of rate of phagocytosis.

Figure 2: Facs staining of Annexin-V (marker for Apoptotic and necrotic cells) and PI (marker for necrotic cells).

Comment#4) The authors write, "Inhibition of CtsK activity also slightly reduced phagosomal proteolysis of peritoneal macrophages... (lines 180-181)." In my opinion, this looks like a significant inhibition (Fig. 3E). The test used was an ANOVA with post-hoc Tukey. Could these authors please add the significance/p value. Given Ctsk was not detected by MS in peritoneal macrophages, this suggests that the Ctsk inhibitor may have non-specific effects. Thus, it would be important to back up the inhibitor data with experiments utilizing Ctsk deficient macrophages (PMID: 15161653).

Response 4: Lung macrophages exhibited increased proteolytic activity that was significantly reduced by CtsK inhibition compared to peritoneal macrophages. MS approach is a very sensitive method, however, we cannot exclude a residual amount of CtsK present in peritoneal phagosomes which was not detected by MS. As described in methods, we use a highly selective CtsK inhibitor, L 006235 ($K_i=0.2$ nM) at concentration $1 \mu\text{M}$ – 1hr pre-treatment. At higher concentration, it can interfere with cathepsin B ($K_i=1 \mu\text{M}$), cathepsin L ($K_i=6 \mu\text{M}$), and cathepsin S ($K_i= 47 \mu\text{M}$). It is therefore possible that the intracellular degradation of BSA mediated by other cathepsins in peritoneal macrophages (Figure 3E), could be partially inhibited by L 006235. Consistently, the effect diminished once collagen, a more specific substrate of CtsK, was used (Figure 3F). We added significance levels to Figure 3E and modified the text accordingly. We appreciate Reviewer's input, we apologize, we believe to get mice with cathepsin K deletion is outside the scope of this study.

Comment#5) Given that TGF-beta lowers Ctsk expression in lung fibroblasts (PMID: 16045809), could it have a dual effect on both enzymatic activity and expression in lung macrophages? Do the authors have experimental evidence that these effects are specific for activity?

Response 5: We thank the Reviewer for the comment. We performed gene expression analysis of Ctsk in lung macrophages treated or not with TGF- β (5 ng/ml) for 12h. The qRT-PCR analysis of Ctsk expression revealed that TGF- β does not significantly affect Ctsk expression in murine lung macrophages (**Supplementary Figure EV4D**). These data indicate that CtsK inhibition occurs on post-translational level, possibly via regulation of activating cleavage of procathepsin K in phagosome for which low pH is required (PMID: 9153258).

Comment#6) The relevance of CD80 expression decreases by TGF-beta in Fig EV3A is not clear in the text.

Response 6: TGF-beta promotes M2-like macrophage phenotype characterized by CD80 down-regulation (PMID: 27418133) as shown by increase expression of Arginase-1 (a classical marker of M2 macrophages) (Figure 3 in *Rebuttal*). We amended the text.

Figure 3: TGF- β induces Arginase-1 (Arg1) expression in lung RTMs. Lung RTMs were stimulated by TGF- β (5, 10 and 20 ng/ml) for 12 h and analysed for Arg1 gene expression. Data information: The statistical significance of data is denoted on graphs by asterisks where **** $p < 0.0001$ or ns = not significant as determined by ANOVA with post-hoc Tukey test. Data are normalized to Arg1 expression in untreated cells.

Comment#7) The methods for isolation of human lung macrophages is not clear. Was the tissue digested? Additionally, no quality control or FACS plots are shown to indicate viability and phenotype of the human cells isolated (PMID: 28769058).

Response 7: We thank the reviewer for the comment. The isolation of human macrophages and all the controls were done as previously described in the paper of our collaborator from AstraZeneca (<https://doi.org/10.1186/s12931-021-01762-4>). Human lung macrophages are highly autofluorescent, the viability of human lung macs were determined by trypan blue assay.

Comment#8) It is not clear if the data from Fig. 4F represents technical replicates or individual donors. Furthermore, it would be important to present the data as RFU in the supplement (similar to Fig. 4B) so the reader can determine the magnitude of delta RFU and easily compare the data across the Figure. Similar point for the data shown in Fig. 4D/E.

Response 8: We thank the reviewer for suggestion. We collected more human patient material and Fig. 4D-G are now presented as individual donors. We understand reviewer's concern and we transformed delta RFU in Fig. 4D-G into proteolytic rates by taking slope of proteolytic signal during the linear phase.

Comment#9) Only 2 patients are shown in Fig. 4G. To strengthen the claim that COPD macrophages exhibit reduced Ctsk dependent collagen degradation versus controls, it would be important to significantly enhance this patient cohort and additionally provide demographics (e.g. sex, age etc between controls and COPD patients).

Response 9: We thank the reviewer. We have added additional two COPD patients (a total number of 4 COPD patients) and compared to all non-COPD samples which were further stratified based on the smoking status. The data demonstrate that alveolar macrophages from COPD patients show reduced efficiency of intracellular degradation of collagen which is at least partially mediated by lower cathepsin K activity. The data are plotted as proteolytic rates by taking slope of proteolytic signal during the linear phase. We provided sex and age information about COPD patients **Table EV2**.

Comment#10) Tumor associated macrophages have been shown to degrade collagen via uptake through the mannose receptor (PMID: 29281816). The authors should cite this paper. Could they speculate on whether the mannose receptor might play a similar role in lung macrophages? Additionally, it would interesting to examine if other M2 stimuli aside from TGF-beta (e.g. IL-4 or IL-10) regulate Ctsk activity as this may strengthen the paper.

Response 10: We thank the reviewer for input and for the reference to the study. We inspected phagosome data for the presence of collagen receptors. Phagosomes of lung RTMs showed the highest abundance of mannose receptor (Mrc1, MR) together with other known collagen receptors (Ly75, Itga1, Itgb1). This suggests lung RTMs are more efficient in phagocytic uptake of collagen which might further aid intracellular collagen clearance as reported by the study suggested by reviewer (PMID: 29281816) (*Figure 4 in Rebuttal*). This study describes up-regulation of cathepsin-endocytosing tumour associated macrophages. Increased activity and aberrant localization of proteases within the tumour microenvironment have a potent role in driving cancer progression, proliferation, invasion and metastasis. Our data demonstrated that both M2-like cytokines, IL-4 and IL-10, significantly decreased Ctsk expression in lung macrophages while TGF-b did not have any effect (*Figure 5 in Rebuttal*). Altogether, these data indicate that IL-4 and IL-10 might have opposing effect in normal lung compared to tumour environment or that other factors, such as macrophage origin contributes to pathology (PMID: 29281816). Therefore, we believe that citation of this paper might be interesting, but it will not add clarity to our study, with all the respect to the Reviewer, we decided not to add it to the paper discussion.

Figure 4: Collagen receptors represent specific components of lung RTMs phagosomes. Heatmap showing relative expression of identified collagen receptors in phagosomes of peritoneal, brain, and lung RTMs. Colour gradient correlates to log-ratio of intensity divided by row median, grey cells indicate missing values. Data information: Data are derived from three replicates.

Figure 5: TGF-β does not significantly alter Ctsk expression in lung RTMs

Lung RTMs were stimulated either by TGF-β (5 ng/ml), IL-4 (20 ng/ml), or IL-10 (20 ng/ml) for 12 h before lysis and qPCR analysis of Ctsk expression.

Data information: The statistical significance of data is denoted on graphs by asterisks where **** p<0.0001 or ns = not significant as determined by ANOVA with post-hoc Tukey test. Data are normalized to Ctsk expression in untreated cells and averaged from four replicates.

Minor points

1) The authors mention that CtsD activity is similar between the cell lysates of both RTMs (Fig. 3C) (lines 174-175). However, pepstatin A is much more effective at inhibiting CtsD in peritoneal than lung macrophages. Can the authors comment on this? Might this reflect expression levels of the different cathepsins (in this case CtsD) between the different RTMs?

Response 1: We thank the reviewer for the comment. Different cathepsin expression might be one of the possible explanation of increased inhibition of cathepsin D in peritoneal macrophages as well as a potential interference of other cathepsins (i.e. pepstatin A-insensitive) in degradation of fluorescence probe in lung macrophages.

2) The authors should discuss the significance of their findings in relation to literature suggesting that during lung fibrosis, fibroblasts and not lung macrophages are the main cells upregulating CtsK and taking up collagen (PMID: 15161653) and that CtsK has been shown to regulate TGF-beta (PMID: 21627832).

Response 2: We thank the reviewer for the input. We amended the discussion.

3) The beta is missing from Fig. 4A

Response 3: We thank the reviewer for the correction. The figure is amended.

December 15, 2022

RE: Life Science Alliance Manuscript #LSA-2022-01535-TR

Dr. Ivo Fabrik
University of Gothenburg, Institute of Biomedicine, Wallenberg Centre for Molecular and Translational Medicine (WCMTM)
Microbiology and Immunology
Medicinaregatan 7A
Biomedical Research Centre, University Hospital Hradec Kralove, Hradec Kralove, Czech Republic
Gothenburg, Sweden/Västra Gotaland 40530
Sweden

Dear Dr. Fabrik,

Thank you for submitting your revised manuscript entitled "Lung macrophages utilize cathepsinK-dependent phagosomal machinery to degrade intracellular collagen". We would be happy to publish your paper in Life Science Alliance pending final revisions necessary to meet our formatting guidelines.

- please address Reviewer 1's final comments
- please upload your figure files as single files and update your supplementary figures in the figure legend section (these should be supplementary figures rather than expanded view figures); please also update the figure callouts in the manuscript text accordingly
- please also rename your EV Tables as supplementary tables and adjust the callouts to the tables in the text accordingly
- please add ORCID ID for corresponding author-you should have received instructions on how to do so
- please use the [10 author names, et al.] format in your references (i.e. limit the author names to the first 10)
- please add callouts for Figure 1B,C and Figure S4D to your main manuscript text
- dataset PXD033010 should be made publicly accessible at this point

A. FINAL FILES:

B. MANUSCRIPT ORGANIZATION AND FORMATTING:

Sincerely,

Reviewer #1 (Comments to the Authors (Required)):

all my concerns are addressed. The authors might seek for language polishing as I found some typos:

- * line 122: "presence for LCCM" should read "presence of LCCM"
- * line 138: "degradation extracellular" should read "degradation of extracellular"
- * line 287: remove comma after the bracket

January 3, 2023

RE: Life Science Alliance Manuscript #LSA-2022-01535-TRR

Dr. Anetta Härtlova
University of Gothenburg
Microbiology and Immunology
Medicinaregatan 7A
Gothenburg, Sweden/Västra Gotaland 40530
Sweden

Dear Dr. Härtlova,

Thank you for submitting your Research Article entitled "Lung macrophages utilize cathepsinK-dependent phagosomal machinery to degrade intracellular collagen". It is a pleasure to let you know that your manuscript is now accepted for publication in Life Science Alliance. Congratulations on this interesting work.

DISTRIBUTION OF MATERIALS:

Again, congratulations on a very nice paper. I hope you found the review process to be constructive and are pleased with how the manuscript was handled editorially. We look forward to future exciting submissions from your lab.

Sincerely,
